# Overweight in Young Athletes: New Predictive Model of Overfat Condition

**DOI:** 10.3390/ijerph16245128

**Published:** 2019-12-16

**Authors:** Gabriele Mascherini, Cristian Petri, Elena Ermini, Vittorio Bini, Piergiuseppe Calà, Giorgio Galanti, Pietro Amedeo Modesti

**Affiliations:** 1Dipartimento di Medicina Sperimentale e Clinica, Università degli Studi di Firenze, 50134 Firenze, Italy; 2Dipartimento di Medicina, Università di Perugia, 06156 Perugia, Italy; 3Sector “Health and Safety in the Workplace and Special Processes in the Field of Prevention”, Directorate of Citizenship Rights and Social Cohesion, 50139 Firenze, Italy

**Keywords:** Triceps skinfold, overweight, youth, obese, child, adolescent

## Abstract

The aim of the study is to establish a simple and low-cost method that, associated with Body Mass Index (BMI), differentiates overweight conditions due to a prevalence of lean mass compared to an excess of fat mass during the evaluation of young athletes. 1046 young athletes (620 male, 426 female) aged between eight and 18 were enrolled. Body composition assessments were performed with anthropometry, circumferences, skinfold, and bioimpedance. Overweight was established with BMI, while overfat was established with the percentage of fat mass: 3.5% were underweight, 72.8% were normal weight, 20.1% were overweight, and 3.5% were obese according to BMI; according to the fat mass, 9.5% were under fat, 63.6% were normal fat, 16.2% were overfat, and 10.8% were obese. Differences in overfat prediction were found using BMI alone or with the addition of the triceps fold (area under the receiver operating characteristics curve (AUC) for BMI = 0.867 vs. AUC for BMI + TRICEPS = 0.955, *p* < 0.001). These results allowed the creation of a model factoring in age, sex, BMI, and triceps fold that could provide the probability that a young overweight athlete is also in an overfat condition. The calculated probability could reduce the risk of error in establishing the correct weight status of young athletes.

## 1. Introduction

The definition of overfat is necessary in order to accurately specify the problem of excess body fat that directly affects health and physical fitness [1]. Excess body fat is a public health problem and is considered an independent risk factor for several non-communicable chronic diseases [2]. Body Mass Index (BMI, expressed in kg/m^2^) is the main parameter in clinical practice of classifying the subject’s weight status. However, the limitation of this method is clearly established. In order to establish the overfat condition, the use of adequate tools for the evaluation of the fat mass (FM) in addition to the BMI value is recommended [3]. This aspect should be considered in particular during the evaluation of an athlete due to greater muscle mass with the same body weight [4].

In the pediatric population, the assessment of weight status is an aspect that the World Health Organization is continuously evaluating, because childhood obesity is associated with a higher chance of obesity, premature death, and disability in adulthood [5]. In addition, for pediatric ages, BMI has low sensitivity to establish excess adiposity, and 25% of children fail to identify their own correct weight status [6]. This difference is particularly evident in the evaluation of young athletes’ obesity [7].

There are numerous techniques and equations in order to quantify FM [8]. The reference methods in this field have some inherent problems, such as the time needed for evaluations, the financial costs involved, or the unnecessary exposure to radiation, in particular with young people [9].

During clinical practice or in sports fields, a simple and low-cost procedure is required to discriminate overweight with low levels of FM from overweight with overfat. For this reason, Junior [10] recommended the use of waist circumferences (WC) and waist-to-height ratio (WTHR) in addition to BMI in order to body fat discriminators in children and adolescents in both sexes.

However, to date it is unknown whether these parameters can be used with the same efficacy even in young athletes. Therefore, the purpose of the present study is to establish a simple and low-cost method that, associated with BMI, provides the discriminating factor between overweight conditions due to a prevalence of lean mass compared to that due to an excess of FM during the evaluation of young athletes.

## 2. Materials and Methods

### 2.1. Subjects

A group of young athletes was enrolled before pre-participation screening for sport eligibility in the Sports Medicine Department. In Italy, the sport eligibility certification is 12 months long but may not coincide with the start of the competitive season; therefore, the athletes were evaluated during the regular season.

We have analyzed data from 1046 young athletes (620 male and 426 female) aged between 8 and 18 years (mean age 13.9 ± 2.4 years for male and 12.7 ± 2.3 years for female). Inclusion criteria for the subjects included being Caucasian, practicing sports at a competitive level, and not having any contraindications to sports eligibility. Exclusion criteria in the analyses were age outside the range of ±6 months compared to the average age of its own stratum, and abnormal values for weight and height (5 kg below the third or 30 kg above the 97th percentile; 5 cm below the third or 5 cm above the 97th percentile) [11].

Written informed consent from both parents or guardians was retrieved for children under the age of 18. After receiving consent, all subjects underwent a voluntary assessment of body composition. The study was carried out in conformity with the ethical standards laid down in the 1975 declaration of Helsinki and was approved by the local ethics committee. This study is part of a project of the Tuscany Region called “Sports Medicine to support regional surveillance systems”; it was approved by the Regional Prevention Plan 2014–2018 with the code O-Range18. Informed consent was obtained from all the participants before inclusion in the study.

### 2.2. Body Composition Assessment

Technical staff trained in data integration of anthropometry, circumferences, skin fold thickness, and bioelectric impedance analysis (BIA) [12] performed total body composition assessments.

Anthropometry and circumferences.

Weight was measured to the nearest 0.1 kg and height to the nearest 0.1 cm (Seca GmbH & Co.). BMI was calculated as body mass divided by height squared (kg/m^2^). Circumference measures were made with a tape metric (Holtain Limited, 1.5 m Flexible Tape) at waist, hip, and biceps:Waist is taken at the narrowest level, or if this is not apparent, at the midpoint between the lowest rib and the top of the hip bone (iliac crest).Hip is taken over minimal clothing at the greatest protrusion of the gluteus muscles. The subject stands erect with their weight evenly distributed on both feet and legs slightly parted without tensing the muscles.Bicep circumference, with the arms relaxed, is taken at the level of the midpoint between the acromion and the olecranon processes.

Skin fold thickness.

Skin fold measurements are widely utilized to assess body FM. Measurements were taken in eight different anatomical sites around the body using calipers (Holtain, Limited Tanner/Whitehouse Skinfold Caliper): triceps, biceps, sub scapula, supra ilium, mid axilla, pectoral, abdominal, and quadriceps (expressed as mm). In addition to the values of the single folds, the sum of all eight folds and the FM percentages derived from skinfold thickness were reported [12,13].

Bioelectric impedance analysis (BIA).

Body impedance is generated in lean tissues as an opposition to the flow of an injected alternate current. Bioelectrical impedance was measured with a phase-sensitive impedance plethysmograph (BIA 101 Sport Edition, Akern, Florence, Italy). The device emits an alternating sinusoidal electric current of 800 mA at an operating single frequency of 50 kHz; standard whole-body tetra polar measurements were performed according to manufacturer guidelines [14]. Resistance (Rz, Ω) is the opposition to the flow of an injected alternating current; reactance (Xc, Ω) is the dielectric or capacitive component of cell membranes and organelles; and phase angle (PA, in degrees) is defined as the ratio between Rz and Xc or between intra and extracellular volumes [15]. In addition, the percentages of FM (FM%) as suggested by McCarthy were recorded [16].

### 2.3. Statistical Analysis

In order to establish underweight, normal weight, overweight, or obese conditions, a subdivision according to BMI, age, and gender was adopted with the International Obesity Task Force (IOTF) guidelines for children and adolescents [17]. In order to establish under fat, normal fat, overfat, or obese conditions, a subdivision according to FM percent, age, and gender was adopted with the body fat reference curves for children developed by McCarthy [16].

The Shapiro–Wilk test was used to assess the normal distribution of variables. The Chi-square test with Yate’s continuity correction was used for comparisons of categorical variables, and the Mann–Whitney’s U-test was used for comparisons of non-normally distributed continuous variables. Multivariate logistic regression models were fitted for the prediction of the dichotomized FM (under fat/normal fat = 0; overfat/obese condition = 1), incorporating to the base model (age, gender, BMI) one at a time in separate models and all the anthropometric variables. To decrease the overfit bias and internally validate our results, all regressions were subjected to 200 bootstrap resamples, and the goodness-of-fit of logistic models were checked using the Hosmer and Lemeshow test. The predictive accuracy of logistic regression models was quantified as the area under the receiver operating characteristics (ROC) curve (AUC), and AUCs were compared using the DeLong method [18]. Odds ratios (ORs) with 95% confidence intervals were also calculated. Furthermore, multivariate logistic regression coefficients were used to develop an FM-based nomogram (Orange software, version 3.4.2, 2017; https://orange.biolab.si/) [19].

All statistical analyses were performed using IBM-SPSS^®^ version 23.0 (IBM Corp., Armonk, NY, USA, 2015). In all analyses, a two-sided *p*-value < 0.05 was considered significant.

## 3. Results

Anthropometrics, skin fold, and bioelectric impedance parameters of the sample with a gender comparison are shown in Table 1.

The prevalence of the weight status of the 1046 subjects depends on which classification is used. Following the IOTF classification (based on BMI) 3.5% were underweight, 72.8% were normal weight, 20.1% were overweight, and 3.5% were obese, compared to 9.5% under fat, 63.6% normal fat, 16.2% overfat, and 10.8% obese, according to the FM. Figure 1 shows the prevalence of the weight status differences between sexes based on the subdivision by BMI or by FM. The division into weight classes carried out by the BMI shows no differences between genders (*p* = 0.375); FM stratification shows a different prevalence (*p* < 0.001), in particular for the overweight condition.

The results of the logistic regression analysis considering the parameters of age, sex, and BMI show a coverage of the AUC equal to 0.867 ± 0.013. Table 2 shows the results of the AUC in the case of the addition of a fourth anthropometric variable to the BMI, age, and gender variables. Triceps, abdominal fold, and skinfold sum have achieved the highest results.

This result allowed the authors to create a model with age, sex, BMI, and triceps fold variables that could provide the probability that an overweight child is also in an overfat condition (Figure 3).

## 4. Discussion

The objective of this study was to provide a practical and simple solution, applicable in a clinical or in a field setting, to differentiate the excess weight due to FM from that due to muscle mass in young athletes.

The review performed by Junior [10] concludes that BMI, waist circumference, and waist-to-height ratio are body fat discriminators in children and adolescents with low operational costs. Another study performed by Sardinha [20] suggests that triceps fold gives the best result in order to define the relationship between BMI and percentage of body fat in healthy boys and girls aged 10 to 15 years, whereas BMI and upper arm girth are reasonable second choices. Both studies suggest specific cutoffs by age group and sex; however, these parameters should be utilized with caution in the evaluation of young athletes [10].

Athletes are subjected to remodeling of the proportions of FM compared to the whole body; these modifications are particularly evident when young athletes are compared to untrained young people [21]. This process occurs due to the different movements and training regimens required by the sport practiced. Therefore, body shape of the athletes could be classified into three categories: muscular and well-balanced type (running, volley, soccer, cycling, and swimming); rich muscular and large-built type (canoeing and rugby); and rich muscular and long-torso type (weight lifting) [22]. Our study enrolled exclusively young athletes. The waist circumference, waist-to-height ratio, and biceps circumference results reached lower statistical power compared to skinfold parameters. Our results suggest using only the triceps fold in order to simplify the assessment: upper arm is easily accessible for detection purposes, and the difference in comparison with abdominal fold (that gave a higher AUC value) is not statistically significant. The evaluation of adiposity in young athletes, unlike non-sporting young people, should include the use of more specific tools that allow a direct assessment of the concerned body compartment.

Currently, the main clinical techniques for the measurement of adiposity in youth are BMI and circumferences, which enable the use of growth centile charts [23]. Skinfold thickness measurement and the use of standardized equations involve some limitations in the estimation of the adiposity in the pediatric population. These biases were particularly high with increasing fatness and were affected by age [24], reaching an error rate of over 5% [25]. Our study did not use any equation for FM estimation but only the triceps fold value [26], expressed in mm.

The prediction model proposed by this study does not provide an absolute certainty of the overfatness condition of a young overweight athlete. However, it provides a probability that further action to promote the health of the young athlete should be considered starting from the value of 50%. Figure 3 describes the procedures that health professionals should follow after pinching the tricep folds of young athletes. This value should be recorded in addition to the age, sex, and BMI values; each value corresponds to a score at the top of the figure. At the bottom of the figure, there is the sum of the scores that provides a probability that the subject is in an overfat condition in addition to overweight.

The skinfold technique requires measurement expertise, and in order to achieve the best results possible, health professionals should undergo training in this technique before carrying it out in order to reach a good reliability and accuracy [27].

Data concerning anthropometrics and body composition, as well as the percentage of body fat, of the present study are in agreement with well-documented sex-related differences in body composition between males and females [28].

The prevalence of overweight and obesity is in line with previous studies [29,30]. However, the weight status analyzed with FM results shows some differences in comparison with BMI results. In particular, the prevalence of the obesity rises from about 3–4% to around 10% in both sexes. In males, the status of overweight is transferred into obesity status, while in females there is a transfer from the state of normal weight in favor of obesity. This differentiation also affects an adequate assessment of cardiorespiratory fitness of young athletes, which in fact is mainly influenced by the FM rather than by the BMI [31]. This is significant because high levels of cardiorespiratory fitness may attenuate the association between excessive adiposity and risks of cardiovascular and metabolic diseases.

The present study shows some strengths. Firstly, the number of subjects that underwent the body composition assessment is in line with those present to date. Secondly, all subjects regularly perform organized sport. Finally, the same expert operator in young athlete evaluation carried out all the anthropometric and skinfold assessments.

The authors are aware that a limitation of the study is the use of BIA to evaluate the percentage of FM, but they followed the study by McCarthy et al. [16] to establish the prevalence of weight status based on FM in a sample of athletes aged eight to 18 of both sexes. This study did not distinguish children and adolescents by maturation phase; it could be inferred that the sample heterogeneity could alter the accuracy of the results. However, the objective of the study does not predict the evaluation following stages of growth, because this aspect involves a further variable that would not allow the creation of a simple tool applicable in a clinical setting or in the field.

This paper should be considered as a preliminary proposal for a new work tool that needs validation on a larger population. At the same time, a future study direction could be extending the possibility of this evaluation to other races as well.

## 5. Conclusions

In summary, obesity during youth is a public health problem that is constantly on the rise worldwide. Participation in organized sports during youth is a widely used activity in western countries because it brings many benefits to children. Among these advantages, there is also weight control. However, it has been shown that BMI has some inherent limitations in assessing adiposity, especially in young athletes. The authors therefore suggest a new predictive model to determine when excess weight is due to excess fat mass. This practice could be used as an additional assessment of young athletes during physical examinations conducted by the sports medicine physicians or physical fitness evaluations conducted by the athletic trainers.

## Figures and Tables

**Figure 1 ijerph-16-05128-f001:**
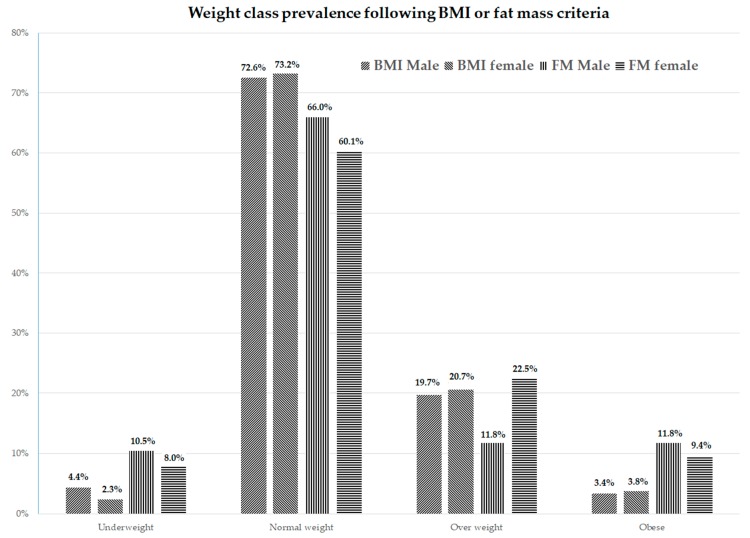
Differences in prevalence weight status between gender in the sample of young athletes based on the subdivision by Body Mass Index (BMI) or by fat mass (FM).

**Figure 2 ijerph-16-05128-f002:**
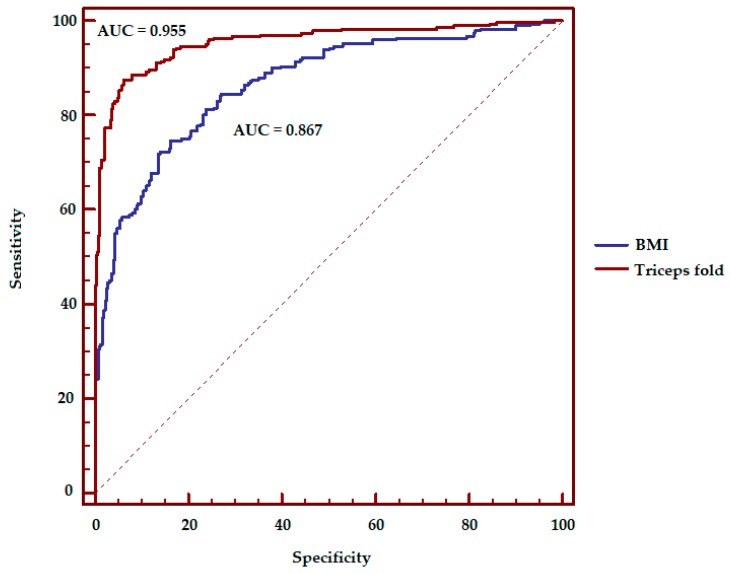
Receiver operating characteristics (ROC) curve, assessment of the accuracy of overfatness prediction using BMI alone (AUC = 0.867) or with the addition of the triceps fold value (AUC = 0.955) in young athletes. Statistical difference were found *p* < 0.001.

**Figure 3 ijerph-16-05128-f003:**
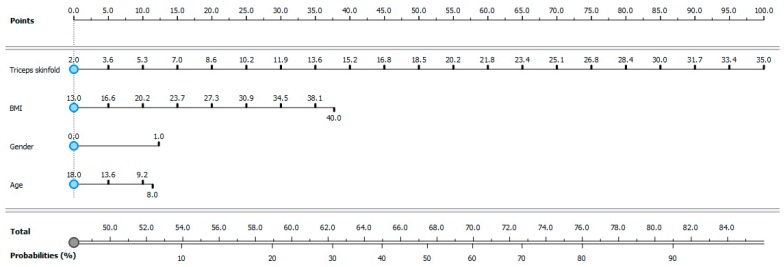
Probabilities that an overweight young athlete is also in overfat condition. Triceps skinfold, BMI, gender, and age are the four variables that give the probability of having an excess of fat mass in addition to the overweight condition. At the top of the figure there is a scores scale corresponding to the values of these four variables. At the bottom of the figure there is a scale that corresponds to the sum of the scores obtained from the four variables (Total) corresponds to a probability (%) that the subject is in an overfat condition. Gender: 0 = Female, 1 = Male.

**Table 1 ijerph-16-05128-t001:** Anthropometrics, skin fold and bioimpedance parameters of the whole sample of young athletes. Data for reading simplicity are expressed as mean ± SD.

Variable	Male (n = 620)	Female (n = 426)	*p* Value
Age (y)	13.87 ± 2.42	12.74 ± 2.33	<0.001
Height (m)	1.64 ± 0.14	1.55 ± 0.12	<0.001
Weight (kg)	55.71 ± 15.24	48.97 ± 13.19	<0.001
BMI (kg/m^2^)	20.23 ± 3.33	20.04 ± 3.33	0.076
Waist circ. (cm)	68.34 ± 8.88	64.22 ± 8.67	<0.001
Hip circ. (cm)	85.18 ± 10.07	83.85 ± 13.41	0.057
Hip/waist	0.807 ± 0.14	0.809 ± 0.46	<0.001
WHR	0.52 ± 0.04	0.54 ± 0.07	<0.001
Biceps Circ. (cm)	24.40 ± 4.30	25.08 ± 29.41	0.001
Biceps fold (mm)	6.07 ± 3.50	8.54 ± 3.87	<0.001
Triceps fold (mm)	11.57 ± 5.48	16.47 ± 5.74	<0.001
Subscapular fold (mm)	9.16 ± 5.18	12.28 ± 8.11	<0.001
Supra iliac fold (mm)	11.08 ± 7.06	15.10 ± 7.37	<0.001
Axilla fold (mm)	8.06 ± 5.25	10.68 ± 6.81	<0.001
Pectoral fold (mm)	7.85 ± 4.96	11.03 ± 5.05	<0.001
Abdomen fold (mm)	12.51 ± 8.02	17.14 ± 7.98	<0.001
Quadriceps fold (mm)	15.29 ± 9.54	22.10 ± 6.51	<0.001
Sum fold (mm)	81.58 ± 42.66	91.25 ± 38.95	<0.001
Fat Mass from Skinfold (%)	16.85 ± 7.22	23.62 ± 7.56	<0.001
RZ (Ω)	566.85 ± 89.97	623.62 ± 69.18	<0.001
XC (Ω)	61.62 ± 8.07	66.22 ± 25.74	<0.001
PA (°)	6.29 ± 0.92	6.08 ± 2.17	<0.001
Fat Mass from BIA (%)	18.25 ± 7.00	25.11 ± 6.34	<0.001
Fat Mass (Kg)	10.43 ± 6.20	12.92 ± 6.35	<0.001
Free Fat Mass (Kg)	45.28 ± 11.79	36.08 ± 7.72	<0.001

**Table 2 ijerph-16-05128-t002:** Area under the curve (area under the receiver operating characteristics curve (AUC) ± SE) results with the addition the fourth anthropometric variable to the BMI, age and gender in young athletes.

Anthropometric Variable	Area under the curve (%)
Waist circumference	0.877 ± 0.011
Waist/height	0.882 ± 0.013
Waist/hip	0.867 ± 0.013
Biceps circumference	0.867 ± 0.013
Triceps fold	0.955 ± 0.008
Sub scapula fold	0.920 ± 0.010
Supra ilium fold	0.938 ± 0.009
Mid axilla fold	0.944 ± 0.009
Pectoral fold	0.946 ± 0.008
Abdominal fold	0.960 ± 0.007
Quadriceps fold	0.938 ± 0.008
Skinfolds sum	0.976 ± 0.006

Differences in overfatness prediction were found using BMI alone or with the addition of the triceps fold value (AUC: 0.867 vs 0.955, *p* < 0.001, Figure 2).

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
