# Peer review of "Overweight in Young Athletes: New Predictive Model of Overfat Condition"

_ijerph, 2019, doi:10.3390/ijerph16245128_

Round 1
Reviewer 1 Report
Comments on manuscript ID: ijerph-658643 “ evaluation of overfat condition in young athletes: a proposal for a new work tool”
Thd aim of this study was to establish a simple method to differentiate overweight conditions due to a prevalent lean mass from an excess of fat mass in young athletes.
I think that it’ s an interesting study; considering that a limitation of the study is the use of BIA for the evaluation of body composition, my only request is to add in table 1 the results of fat mass % evaluated both by bioimpedance analysis and skinfold thickness in order to better understand the accuracy of BIA.
Author Response
Thd aim of this study was to establish a simple method to differentiate overweight conditions due to a prevalent lean mass from an excess of fat mass in young athletes.
I think that it’ s an interesting study; considering that a limitation of the study is the use of BIA for the evaluation of body composition, my only request is to add in table 1 the results of fat mass % evaluated both by bioimpedance analysis and skinfold thickness in order to better understand the accuracy of BIA.
Dear reviewer
Thank you for the work done in order to improve our manuscript.
You ask to add FM% value derived from skinfold in table 1.
The authors add these values keeping the same style of the table.
The value is on average lower in both males and females and the difference between the sexes is confirmed.
A sentence has also been modified in the methods section " In addition to the values of the single folds, the sum of all eight folds and the FM percentages derived from skinfold thickness were reported [12, 13].]”

Reviewer 2 Report
The authors present a novel method "new work tool" to detect over-fat in young athletic subjects. Strengths of the study are that a large cohort was included and that this study address a pressing problem in society. As the authors phrase it "obesity during youth is a public health problem". There are, however, several changes that are needed in the paper.
Major concerns:
(1) Proof reading by native English speaker is required.
(2) Ethic statement is missing important information who gave consent, please state that "consent from both parents or guardians were retrieved for children under the age of 18".
(3) Validation for the "new work tool" is missing, of course the probability will be accurate (Figure 3) since the same cohort is used to describe the probabilities as the model was developed using. In order to give credibility to the suggested method the validation must be performed in a second cohort. Suggest to divide the cohort in two and use the first half to develop the model and the second half to validate the method. The statement "Figure 3 describe the procedure that health professionals should follow after pinch the triceps fold of an young athletes." does not follow from the results since the suggested method is not validated.
(4) The statement in the introduction "...to date there is no universally applicable criterion for body composition assessment [9]" is not accurate. This is an old reference (from 2012) and since then several methods have been developed and today there exists gold standard methods for body composition assessment, such as MRI or DXA. The argument the authors could instead make is that these methods are costly and challenging in larger cohorts.
Minor concerns:
(1) Why was only Caucasians eligible for inclusion? Please motivate why this selection was made.
(2) Figure 1 is missing labels on x-axis, if this is only for the groups please use a box-diagram instead as this suggest a continuous scale.
(3) Figure 3 is challenging to understand please provide a more thorough explanation how to interpret and use?
Author Response
The authors present a novel method "new work tool" to detect over-fat in young athletic subjects. Strengths of the study are that a large cohort was included and that this study address a pressing problem in society. As the authors phrase it "obesity during youth is a public health problem". There are, however, several changes that are needed in the paper.
The authors would like to thank the reviewer for the work done to improve the manuscript.
In addition, I am proud that the manuscript has been appreciated.
Major concerns:
(1) Proof reading by native English speaker is required.
The English language was corrected by a native editing service (certificate is upload)
(2) Ethic statement is missing important information who gave consent, please state that "consent from both parents or guardians were retrieved for children under the age of 18".
The sentence has been modified:
"Written informed consent from both parents or guardians was retrieved for children under the age of 18. After receiving consent, all subjects underwent a voluntary assessment of body composition."
(3) Validation for the "new work tool" is missing, of course the probability will be accurate (Figure 3) since the same cohort is used to describe the probabilities as the model was developed using. In order to give credibility to the suggested method the validation must be performed in a second cohort. Suggest to divide the cohort in two and use the first half to develop the model and the second half to validate the method. The statement "Figure 3 describe the procedure that health professionals should follow after pinch the triceps fold of an young athletes." does not follow from the results since the suggested method is not validated.
This authors agree with the reviewer.
The paper is focused to test the ability of some anthropometric variables to predict the overfat condition. The analysis takes into account the probabilities of being case or control derived from all the variables in the full models and, as reported in statistical analysis a check of internal calibration (predicted vs observed) for each model has been performed.
Hosmer and Lemeshow test never showed a significant difference between observed and expected frequencies, this means that the hypothesized models well explained the available data. In addition, the limited number of subjects in some age ranges and the disequilibrium between sex prevented us from further sub stratifying the current population into different subgroups as would be required to create a calibration plot. However, as suggested by the Referee, external validation of a predictive tool should be mandatory to establish whether the tool works satisfactorily in different patient population. External validation of our findings within larger population will allow us to better determine discriminative and calibration properties of the prediction model and we are currently working on this. Therefore this paper remain a preliminary proposal for a new work tool yet to be externally validated.
For these reasons the title has been changed to:
"Overweight in young athletes: new predictive model of overfat condition".
In addition the word "tool" has been replaced with "predictive model" in the abstract and in the discussion section.
A new sentence has been add at the end of the discussion:
“This paper should be considered as a preliminary proposal for a new work tool that needs validation on a larger population.”
(4) The statement in the introduction "...to date there is no universally applicable criterion for body composition assessment [9]" is not accurate. This is an old reference (from 2012) and since then several methods have been developed and today there exists gold standard methods for body composition assessment, such as MRI or DXA. The argument the authors could instead make is that these methods are costly and challenging in larger cohorts.
The authors are agree with the reviewer. The sentence has been modified:
"There are numerous techniques and equations in order to quantify FM [8]. The reference methods in this field have some inherent problems, such as the time needed for evaluations, the financial costs involved, or the unnecessary exposure to radiation, in particular with young people [9]."
Minor concerns:
(1) Why was only Caucasians eligible for inclusion? Please motivate why this selection was made.
The authors understand the reviewer's observation.
Body composition is influenced by race.
The Caucasian race in Italy is the predominant one. For this first study the authors chose the Caucasian race in the inclusion criteria to reach a large sample size. However, extending this calculation of probability in other races could be a new direction of future study. A new sentence has been added to the end of the discussion section.
"At the same time, a future study direction could be extending the possibility of this evaluation to other races as well "
(2) Figure 1 is missing labels on x-axis, if this is only for the groups please use a box-diagram instead as this suggest a continuous scale.
As the reviewer suggested figure 1 has been modified in histograms.
(3) Figure 3 is challenging to understand please provide a more thorough explanation how to interpret and use?
Thank you for this observation. The authors has been added "Triceps skinfold, BMI, gender and age are the four variables that give the probability of having an excess of fat mass in addition to the overweight condition. At the top of the figure there is a scores scale corresponding to the values of these four variables. At the bottom of the figure there is a scale that corresponds to the sum of the scores obtained from the four variables (Total) corresponds to a probability (%) that the subject is in a overfat condition." in figure 3 legend.
In addition to the explanation in discussion section: ” Figure 3 describes the procedures that health professionals should follow after pinching the triceps fold of young athletes. This value should be recorded in addition to the age, sex, and BMI values; each value corresponds to a score at the top of the figure. At the bottom of the figure, there is the sum of the scores that provides a probability that the subject is in an overfatness condition in addition to overweight.”

Round 2
Reviewer 2 Report
Thank you for adequately addressing the comments and concerns.
One minor final comment is to use "ethnic groups" instead of "races" as races may be outdated.